# SARS-CoV-2 and Swabs: Disease Severity and the Numbers of Cycles of Gene Amplification, Single Center Experience

**DOI:** 10.3390/children10050841

**Published:** 2023-05-06

**Authors:** Raffaele Falsaperla, Vincenzo Sortino, Ausilia Desiree Collotta, Silvia Marino, Piero Pavone, Laura Grassi, Grete Francesca Privitera, Martino Ruggieri

**Affiliations:** 1Neonatal Intensive Care Unit and Neonatal Accompaniment Unit, Azienda Ospedaliero-Universitaria Policlinico, “Rodolico-San Marco”, San Marco Hospital, University of Catania, 95121 Catania, Italy; 2Unit of Clinical Paediatrics, Azienda Ospedaliero-Universitaria Policlinico, “Rodolico-San Marco”, San Marco Hospital, 95121 Catania, Italy; 3Postgraduate Training Program in Pediatrics, Department of Clinical and Experimental Medicine, University of Catania, 95123 Catania, Italy; 4Unit of Clinical Pediatrics, Azienda Ospedaliero-Universitaria Policlinico, PO “G. Rodolico”, University of Catania, 95123 Catania, Italy; 5Bioinformatics Unit, Department of Clinical and Experimental Medicine, University of Catania, 95123 Catania, Italy; 6Unit of Rare Diseases of the Nervous System in Childhood, Department of Clinical and Experimental Medicine, Section of Pediatrics and Child Neuropsychiatry, Azienda Ospedaliero-Universitaria Policlinico, PO “G. Rodolico”, University of Catania, Via S. Sofia, 87, 95123 Catania, Italy

**Keywords:** SARS-CoV-2 infection, COVID-19, emergency, pediatric care, PCR amplification, LqSOFA score

## Abstract

Pediatric COVID-19 determines a mild clinical picture, but few data have been published about the correlation between disease severity and PCR amplification cycles of SARS-CoV-2 from respiratory samples. This correlation is clinically important because it permits the stratification of patients in relation to their risk of developing a serious disease. Therefore, the primary endpoint of this study was to establish whether disease severity at the onset, when evaluated with a LqSOFA score, correlated with the gene amplification of SARS-CoV-2. LqSOFA score, also named the Liverpool quick Sequential Organ Failure Assessment, is a pediatric score that indicates the severity of illness with a range from 0 to 4 that incorporates age-adjusted heart rate, respiratory rate, capillary refill and consciousness level (AVPU). The secondary endpoint was to determine if this score could predict the days of duration for symptoms and positive swabs. Our study included 124 patients aged between 0 and 18 years. The LqSOFA score was negatively correlated with the number of PCR amplification cycles, but this was not significant (Pearson’s index −0.14, *p*-value 0.13). Instead, the correlation between the LqSOFA score and the duration of symptoms was positively related and statistically significant (Pearson’s index 0.20, *p*-value 0.02), such as the correlation between the LqSOFA score and the duration of a positive swab (Pearson’s index 0.40, *p*-value < 0.01). So, the LqSOFA score upon admission may predict the duration of symptoms and positive swabs; the PCR amplification of SARS-CoV-2 appears not to play a key role at onset in the prediction of disease severity.

## 1. Introduction

Coronavirus disease is caused by Coronavirus 2, which is responsible for a severe acute respiratory syndrome (SARS-CoV-2) that has rapidly spread from China to other States of the World since December 2019 [1], determining a global pandemic from March 2020 [2]. This disease caused about 760 million confirmed cases, with about 6.9 million deaths in the world [3]. Data from the literature show that children are commonly less affected than patients of any other age [4,5,6]. Children, in fact, often present a mild clinical picture characterized by mild respiratory symptoms, fever, a dry cough, fatigue, intestinal alterations or are totally asymptomatic. Of interest, the infections caused by other types of Coronavirus (non-SARS-CoV-2) are also common, recurrent, and frequently asymptomatic in pediatric age [7]. Upon admission to the hospital, patients are clinically evaluated, and a molecular nasopharyngeal swab for SARS-CoV-2 is performed as the gold standard for the diagnosis of SARS-CoV-2 infection. Literature data on the clinical utility of PCR amplification cycles [8] show the correlation between the number of PCR amplification cycles of SARS-CoV-2, usually named the Ct value (Cycles threshold), analyzed from respiratory samples, and disease severity. In fact, lower Ct values were associated with more severe disease, but few data were published on pediatric patients [8]. In the Department of Pediatrics Emergency and Urgency of University Hospital “G.Rodolico-San Marco” of Catania (Italy), doctors clinically evaluated each patient using a Liverpool quick Sequential Organ Failure Assessment, also named the LqSOFA score (range 0–4), which permitted an easy and rapid assessment using the following clinical parameters recorded during the early stages of medical examination: age-adjusted heart rate, respiratory rate, capillary refill, and consciousness level evaluated with the AVPU scale [9]. 

The aim of our study was to establish whether disease severity could be associated with the gene amplification of SARS-CoV-2 obtained by positive molecular nasopharyngeal swabs performed upon admission to the hospital. Thus, the authors studied the correlations between disease severity evaluated during hospital admissions through the LqSOFA score and the number of cycles of the gene amplification of the SARS-CoV-2 gene target N2 as a primary endpoint. In this way, the authors proposed the establishment of a clinical value of the number of cycles of SARS-CoV-2 gene N2 amplification to determine whether this information could be useful to the emergency room physician in establishing the prognosis of each patient.

The secondary endpoints were the correlations between the LqSOFA score assessed during hospital admission and positive swabs duration and the duration of symptoms to study if this score assigned by the time of admission could be used to predict the course of the disease. Finally, we also studied the relationship between the LqSOFA score and the age of patients in order to understand which patients were at a higher risk and needed more attention during clinical and instrumental evaluation. 

Knowledge of this information is essential to minimize the isolation of pediatric patients: a condition that causes serious psychological and social repercussions for the patient and his family.

## 2. Materials and Methods

We performed a retrospective single-center cohort study from April 2020 to March 2022. This study conformed to the Strengthening the Reporting of Observational Studies in Epidemiology (STROBE) Statement, where all points of the checklist have been respected.

### 2.1. Sampling Strategy

In the beginning, we collected all positive molecular swabs in patients between 0 days of life and 18 years of age, accessing the database of the Microbiology laboratory of San Marco University Hospital of Catania (Italy). For each sample, we analyzed the Ct value (threshold value) of the gene N2 of SARS-CoV-2.

For each name, we traced the emergency room reports and any medical records concerning that patient. Therefore, we evaluated the severity of urgency and severity of disease upon hospital admission using the LqSOFA score [8], clinical data at admission, the discharge diagnosis, if respiratory support was started, and which support was used.

To investigate all clinical and epidemiological aspects of pediatric COVID-19, we performed a telephone interview with a semi-structured questionnaire regarding: (1)The number of days with positive swabs;(2)Symptoms of COVID-19;(3)The duration of the symptoms.

### 2.2. Inclusion and Exclusion Criteria

The inclusion criteria of our study were: (1)Positive swabs with an available PCR Ct value amplification of gene N2;(2)If the parent’s telephone number was available for a telephone interview with the researchers;(3)If the phone call with the researchers was answered;

The exclusion criteria were: (1)Patients with incomplete clinical information;(2)Patients with no available PCR Ct gene N2 amplification result;(3)Patients with no correct telephone number on the emergency database;(4)Patients who did not answer the telephone call of researchers:

Written informed consent was obtained from all parents of the patients. 

### 2.3. Population Characteristic

We interviewed 143 parents of the patients by phone and submitted them to the following semi-structured questionnaire: (1)The number of days with positive swabs;(2)Symptoms of COVID-19;(3)The duration of the symptoms;

We also studied all medical records to collect the following clinical data: (1)Severity at the time of hospital admission (evaluated with LqSOFA score);(2)Clinical information on hospitalization;(3)If respiratory support was started and which support was used;

In our study, we found that in 90% of cases, the mother of the patient answered the phone call, and in 10%, the father replied to the medical questionnaire. The average telephone call duration was 5 min and 15 s. 

The questionnaire replies were interested in 132 cases (92.31%), not interested in 10 (6.99%), did not understand the language of the interviewer in 1 (0.7%), and were interested but did not remember the duration of a positive swab or the clinical picture of COVID-19 in 8 cases (6.06%). 

So, in the end, patients that were included in the study for statistical analysis were 124, including 61 males (49%) and 63 females (51%). The average age of the patients was 4.42 years (from 3 days to 15 years).

### 2.4. LqSOFA Score

A modified score of qSOFA, called the LqSOFA score, was proposed for the first time in a retrospective study published in 2020 and conducted at Alder Hey Children’s Hospital in Liverpool [9].

The LqSOFA score (range, 0–4) incorporates age-adjusted heart rate, respiratory rate, capillary refill and consciousness level on the Alert, Voice, Pain, Unresponsive scale (AVPU scale). For each variable considered, a value of 0 or 1 was assigned, which was subsequently summed to obtain the LqSOFA score as reported in Table 1.

In this large study, the authors demonstrated improved performance of the LqSOFA score over the qSOFA score (and other scores such as PEWS and NICE) in identifying febrile children at risk of critical care admission and sepsis-related mortality. This study was performed by comparing the validity of these scores when predicting admission to the intensive care unit in a cohort of critically ill patients, and the LqSOFA proved to be the score with the greatest predictive power in the pediatric population. 

With a cutoff ≥2, this score revealed excellent specificity but was limited by relatively low sensitivity. However, a cutoff of ≥1 for LqSOFA resulted in a more favorable balance of sensitivity and specificity. Therefore, we considered a score greater than or equal to 1 as a cut-off and an indicator of disease severity during admission.

### 2.5. Laboratory Analysis of Swabs

Our laboratory used the “Xpert Xpress SARS-CoV-2 test” developed by Cepheid, a real-time RT-PCR test for the qualitative detection of the nucleic acid of SARS-CoV-2 in upper respiratory samples (such as nasopharyngeal, oropharyngeal, nasal, mid-turbinate swab or nasal wash or aspirate) [11]. According to current scientific evidence, Ct values greater than or equal to 40 was considered negative. Ct values less than 40 were positive [12,13].

This analysis was made using primers and probes in relation to 110,206 SARS-CoV-2 sequences, which were available at the date of 21 October 2020, in the GISAID gene database for only two genetic targets of the viruses, E and N2 [11]. 

GeneXpert instrumental systems automatized and integrated sample preparation, nucleic acid extraction and the amplification and detection of target sequences by real-time PCR assays. This test could provide rapid detection of pandemic SARS-CoV-2 at 30 min in positive samples and in about 45 min in negative specimens. The time of sample preparation by the operator was less than one minute. 

Xpert Xpress SARS-CoV-2 test provides test results based on the detection of two gene targets, E2 and N2. In this way, the molecular test showed the presence of both viral targets in a positive result, one of the two in a presumptive positive result, or none of the two in a negative result. The test was presumed positive when SARS-CoV-2 nucleic acids were present, and, in this case, the test had to be repeated. In detail, the test results were positive when both SARS-CoV-2 target nucleic acids were detected in the sample. Of interest, beyond these two genes, there were many other genes that, unfortunately, our laboratory did not test. The SARS-CoV-2 genome consists, in fact, of many genes. Structural protein genes encode the spike protein (S), membrane protein (M), virion coat protein and nucleocapsid protein (N). Genes for non-structural proteins, on the other hand, which code for proteins, such as RNA-dependent RNA polymerase (RdRp), helicases, exo-endonucleases and accessory proteins, have a role that is not yet fully understood.

Considering the pathogenicity of SARS-CoV-2 and the trend in the diffusion of SARS-CoV-2 to ensure the reliability of the test, in the United States, for example, as many as 4 SARS-CoV-2 genes were detected (N1, N2 and N3 and the human RNase P gene) [12].

### 2.6. Endpoint

In correlation to our objective, we selected the LqSOFA score evaluated during hospital admission compared with the number of amplification cycles of the SARS-CoV-2 gene N2 as the primary endpoint (also definable as the true endpoint).

Furthermore, we asked ourselves what could help doctors to understand which patients could have a worse outcome. Therefore, we selected as secondary endpoints (also definable as surrogate endpoints) the LqSOFA score when evaluated upon hospital admission compared with the duration of a positive swab and COVID-19 symptoms. We also studied the interrelationship between the LqSOFA score upon hospital admission and the age of the patient to study the influence of age on disease severity.

### 2.7. Statistical Analysis

All statistical analyses were conducted using R (v.4.2.1). In particular, we used a Pearson correlation analysis to compare the quantitative variables. 

Pearson’s correlation method is the most commonly used method for numerical variables. It gives a value between −1 (total negative correlation) and 1 (total positive correlation). A positive correlation between two variables states that they grow or decrease together, while a negative one denotes the opposite behavior between the two variables. 

The correlation coefficient (R) was calculated using the data of 124 pediatric patients infected with SARS-CoV-2. In particular, the variables compared included LqSOFA score, age in months, COVID-19 positivity duration in days, symptom duration in days and the number of PCR cycles (Ct values) to detect the virus gene N2. Correlation coefficients more than 0.7 indicated a strong correlation, correlation coefficients between 0.69 and 0.10 indicated a moderate/low correlation, while correlation coefficients under 0.10 indicated a negligible correlation. The R-value was considered significant only with an associated *p* value < 0.05. Each correlation was plotted via the package ggpubr (v.0.6.0) with a scatter plot to show the linear relationship.

## 3. Results

During the study period, 5165 pediatric nasopharyngeal molecular swabs were performed by our microbiology laboratory. Of these, 612 were positive (12%), and 4553 were negative (88%) for SARS-CoV-2 infection. Data on the PCR amplification of the target gene N2 of SARS-CoV-2 was available for 267 positive pediatric molecular swabs (44% of total positive swabs), and of these, only 143 records (54%) fully met the criteria for inclusion in the study (Figure 1).

### 3.1. Severity of Disease at the Hospital Admission

The severity of the disease was assessed using the LqSOFA score during the patient’s stay in the hospital and was, on average, 1, with values between 0 (asymptomatic patients) and 4 (critical patients). 

The distribution of LqSOFA in our sample is reported below:Twenty-four patients with LqSOFA of 0 (19.35%)Seventy-four patients with LqSOFA of 1 (59.68%)Sixteen patients with LqSOFA of 2 (12.90%)Nine patients with LqSOFA of 3 (7.26%)One patient with LqSOFA of 4 (0.81%)

There were 100 hospitalized patients (81%), while 24 (19%) were cared for at home. 

Of interest, only 11 hospitalized patients (8.8%) needed respiratory support, which was invasive in two patients (1.6%) and non-invasive in the remaining nine (7.2%). 

### 3.2. Duration of Positive Molecular Swab and Symptoms Duration

Molecular swabs for SARS-CoV-2 had a positive result for the mean duration of 14.5 days, with values ranging between 5 days and 42 days. Patients in our study showed symptoms for a mean duration of 6.5 days with a range of values between 0 days (asymptomatic patients) and 51 days.

### 3.3. Symptoms of COVID-19 in Our Sample

The most common symptoms in our sample were: fever (69.3%), cough (28.2%), nervous system involvement (21.8%), gastrointestinal changes (14.5%) and dyspnea (8.1%). There were only eight asymptomatic patients (6.45%). Symptoms of the nervous system were asthenia (25.9%), seizures (18.6%), headache (14.8%), anosmia/ageusia (14.8%), sensory alterations (11.1%), dizziness (7.4%), hyposthenia of the limbs (3.7%), drowsiness (3.7%). Seizures were always associated with fever except in one case that occurred without fever but in a patient with a chromosomic disorder.

### 3.4. Statistical Correlations

The first correlation that was studied was between the LqSOFA score assessed at the time of hospital admission and PCR cycle numbers of the SARS-CoV-2 gene N2 amplification (Figure 2A) with a weak negative correlation between these variables (Pearson’s index −0.14) though this was not statistically significant (*p* value 0.128). 

The second correlation that was studied was between the LqSOFA score evaluated during hospital admission and age (in months) (Figure 2B), with a weak negative correlation between these variables (Pearson’s index −0.23), which was statistically significant (*p* value 0.009).

Therefore, to determine the prognosis based on clinical severity evaluated at the onset of disease, the authors decided to study, according to secondary endpoints, whether there was a correlation between the LqSOFA score assessed at the time of hospital admission, the duration of symptoms and the duration of a positive swab. In the first case, there was a moderate correlation (Pearson’s index 0.4) that was statistically significant (*p* value < 0.001) (Figure 2C). In the second case, there was a weak positive correlation (Pearson’s index 0.2) that was statistically significant (*p* value 0.023) (Figure 2D).

The results of the correlations are summarized in Table 2. In Table 3, we report the average symptom duration with a standard deviation for each level of the LqSOFA score.

## 4. Discussion

In the patients included in our study, from April 2020 to March 2022, we found a weak negative correlation between the LqSOFA score assessed at the time of hospital admission and the number of PCR amplifications of the N2 gene of SARS-CoV-2, which may suggest that children with q high viral load demonstrated by lower Ct values had greater severity of disease, even if this correlation was not statistically significant in our population (*p* value 0.128). 

Similar studies in the literature indicate the correlation between a lower number of PCR amplifications, which point out a high viral load, and disease severity, including adults and children [14,15,16,17,18]. Huang et al. in their retrospective study on 308 adults, showed that the viral load of critical patients was more than severe compared to general patients, where this decreased with pharmacologic treatment [14]. Liu et al. analyzed viral dynamics in 76 mild and severe cases of COVID-19 and noted that Ct values of severe cases were significantly lower than mild cases at the time of admission [15]. This correlation was present in another study by Liu et al., including 12 patients, but of these, only one was a pediatric patient of 10 years; in these patients, the Ct value was reciprocally correlated to disease severity [16]. Xia et al. also demonstrated, in their case series, that RT-PCR-CT values in patients with severe-type infection were lower than those in the rest, indicating a correlation with the severity of the disease at an early stage. Of interest, lower RT-PCR-CT values also indicated greater infectivity in the patient [17]. Yu et al. analyzed the correlation between the viral load of SARS-CoV-2 in the sputum of 92 adult patients and the risk of COVID-19 progression: severe patients had significantly lower Ct values during hospital admission than mild-moderate cases with a positive association between their sputum viral load and disease severity (*p* = 0.017) [18]. Of interest, an important Italian study reported a significant correlation between serum IgG titers measured after 30 days from diagnosis and the Ct value of gene N2 performed at the onset of the disease. The highest levels of antibodies in this group of children were due to a higher viral load [19]. 

Our data confirm the correlations found in the literature between the number of cycles of SARS-CoV-2 genome amplification and disease severity, although, in our sample, this result was not statistically significant. Therefore, the authors believe that further large-scale studies are needed to confirm this finding in pediatric patients.

In our patients, the LqSOFA score evaluated during hospital admission was inversely related to age and was statistically significant (*p* value < 0.05), perhaps suggesting that this inverse correlation was due to the indirect effect of disease (such as dehydration, feeding difficulty or gastrointestinal disorder, especially diarrhea) which is more frequent in younger children with an aggravation of clinical conditions. For this reason, it is essential to carefully evaluate all patients of a minor age who present to the emergency department, paying particular attention to the concomitant presence of non-respiratory symptoms such as feeding difficulties or bowel disturbances. Therefore, viral co-infections that are very frequent in infants and children may play an important role in the replication and pathogenetic action of SARS-CoV-2 by direct virus–to–virus interactions and competition [20]. 

According to the secondary endpoints, we studied the relationship between the LqSOFA score assessed at the time of hospital admission, the duration of symptoms and how long swabs remained positive. In both cases, the evaluation of the LqSOFA score upon admission was directly correlated with other variables: this score could predict, with statistically significant results (*p* value < 0.05), the duration of symptoms and positive swabs. At the time of writing, no other studies in the literature are present about this correlation.

In this way, clinicians can show caregivers an estimation of how long their children will remain symptomatic and how the long swabs will remain positive, estimating the period during which children may need to remain in isolation at home to avoid the contagion of other people. In fact, an important study on the viral dynamics of COVID-19 confirmed this hypothesis because the authors showed that patients with a milder clinical picture presented an early viral clearance with a negative molecular nasopharyngeal swab on day 10 after the onset [15]. 

These data are very useful for clinicians because they can use the LqSOFA score both to stratify the risk of needing intensive care and to predict the course of symptomatic and asymptomatic disease in patients with only a positive molecular swab analyzed with a rapid laboratory method and without symptoms. In this way, it is also possible to avoid delays in medical specialist visits or the clinical follow-up of chronic conditions. 

In fact, during the first pandemic phase, in Italy, there was a reduction in hospitalizations and outpatient visits in almost all pediatric areas (especially in the medical area) due to fear of infection, but there was an increase in children evaluated for severe illnesses and the percentage of patients hospitalized for serious diseases increased [21].

COVID-19 in children may have respiratory, neurological, gastrointestinal, or cutaneous presentations, either in combination with other presentations or alone [22]. In pediatric age, COVID-19 can present five clinical patterns: asymptomatic, mild (with mild clinical symptoms such as fever, fatigue, cough, anorexia, malaise, muscle pain, sore throat, nasal congestion and headache), moderate (with moderate clinical symptoms including mild pneumonia manifestation), severe (patients that develop more-severe pulmonary symptoms including dyspnea, central cyanosis and hypoxia) and critical (including patients with MIS-C (Multisystem Inflammatory Syndrome in Children) with acute respiratory failure, shock, and or multiorgan dysfunction). Fortunately, the clinical presentation of SARS-CoV-2-related disease is rare in pediatrics [6]. Our data support this statement. In fact, only 1.6% of our patients presented a critical condition requiring invasive respiratory support.

Regarding the clinical features, in our sample, the most common symptoms were fever (69.3%), cough (28.2%), nervous system involvement (21.8%), gastrointestinal changes (14.5%) and dyspnea (8.1%) and these results are similar to the most common symptoms described in the literature [22].

The indications of our study are very important in deciding when is opportune to repeat the molecular swabs to certify a negative condition, optimize economic resources and avoid excessive periods of isolation that can create serious medical, social, and psychological consequences for families and children.

## 5. Limitations of the Study

Our study has some limitations.

First, it was performed on a small cohort of patients, with a retrospective analysis in part based on a clinical verbal interview with the parents.

In addition, the cohort studied was very variable because the study period included vaccinated and non-vaccinated children and both children who required hospitalization and those that did not, including patients of a neonatal age (0–28 days).

Finally, in our study, no patient had lung CT scans to stratify the severity of respiratory disease from a radiological point of view.

## 6. Conclusions

A molecular SARS-CoV-2 nasopharyngeal swab with the PCR amplification of the genes target, resulting in a positive or negative result, is the base for a microbiological diagnosis of infection. The severity of the disease may be correlated with the number of PCR amplification cycles of the SARS-CoV-2 gene N2, but in our sample, this correlation was not strong or statistically significant.

Therefore, it is necessary to conduct other large-scale studies to define the exact clinical role of the PCR amplification of the SARS-CoV-2 genome in pediatric patients, including multicenter studies focusing on viral dynamics.

Of interest, the use of the LqSOFA score to assess the severity of disease during hospital admission should be promoted both for its ease of use and for its good degree of predictiveness toward the duration of symptoms and positive swabs though other studies with larger samples are necessary to confirm these data. In this way, doctors can predict the course of disease in all patients, including those who have been diagnosed with SARS-CoV-2 infection, by qualitative rapid methods saving time and economic resources.

## 7. Declaration

### Consent to Participate and for Pubblication

Each parent expressed their consent to participate in the study, and the publication of the data took place in an anonymous form. 

## Figures and Tables

**Figure 1 children-10-00841-f001:**
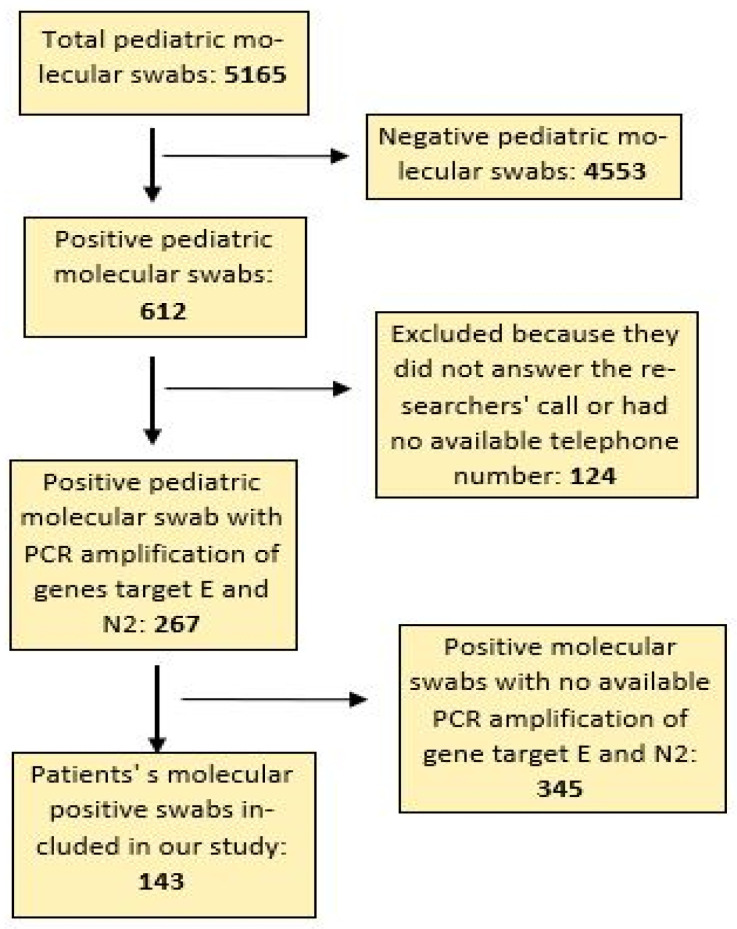
Patients enrolled in the study.

**Figure 2 children-10-00841-f002:**
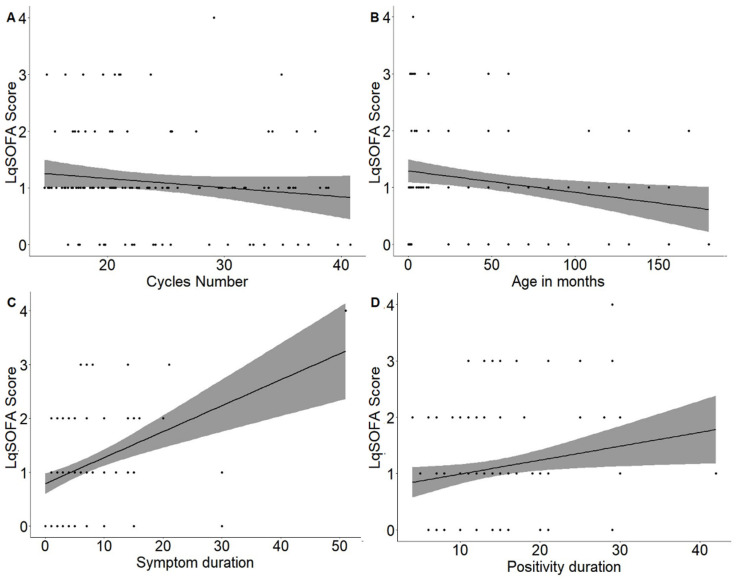
Correlation between LqSOFA score and number of PCR cycles (**A**), age (in months) of the patient (**B**), duration of symptoms (**C**) and duration of positive swab in days (**D**).

**Table 1 children-10-00841-t001:** LqSOFA score.

Point	Capillary Refill Time	AVPU Scale	Heart Rate	Respiratory Rate
1 point	>3 s	V-P-U	>99th centile Bonafide et al. age-specific thresholds [10]	>99th centile Bonafide et al. age-specific thresholds [10]
0 point	<3 s	A	<99th centile Bonafide et al. age-specific thresholds [10]	<99th centile Bonafide et al. age-specific thresholds [10]

**Table 2 children-10-00841-t002:** Correlations between LqSOFA score and positivity duration, symptom duration, age in months and cycle number.

		Pearson Correlation	*p* Value
LqSOFA	Positivity duration	0.20	0.02
	Symptom duration	0.40	<0.01
	Age in months	−0.23	<0.01
	Cycles number	−0.14	0.13

**Table 3 children-10-00841-t003:** Distribution of symptoms duration (days) for each LqSOFA score.

		Symptoms Duration (Days)		
LqSOFA	Patients Number	Mean	Standard Deviation	Median
0	24	3.92	6.65	1.50
1	74	6.00	4.67	5
2	16	8.37	6.71	6
3	9	9.44	4.93	7
4	1	51.00	NA	51

## Data Availability

The datasets used and analyzed during the current study are available from the corresponding author upon reasonable request.

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
