# Peer review of "SARS-CoV-2 and Swabs: Disease Severity and the Numbers of Cycles of Gene Amplification, Single Center Experience"

_children, 2023, doi:10.3390/children10050841_

Round 1
Reviewer 1 Report
In title: since it is a bit long, authors could consider deleting “correlationship between”.
In abstract: review format (bold words).
In keywords: always fitting to this journal guidelines, if posible, include more keywords so as to boost the visibility of this paper in the different databases this journal is indexed in.
In introduction: is it posible to update the information about the quote number 3? Change “In our institute” for its real name (avoiding the inclusion of “our”). Try to develop the gap of knowledge this study aims to cover as well as its importance.
In material and methods: write Liverpool quick Sequential Organ Failure Assessment the first time authors write LqSOFA and, the rest, only LqSOFA. Review if the tables and figures format fits the journal’s guidelines.
In results: Figure 1 has two titles. Figure 2 does not have a title. The text under Figure 2 does not have the same format that the rest of the text. Review the spaces between figures and text.
In discussion: develop the part where you write about this study implications. Try to explain these results in depth based on previous published literature. When were data collected?
In conclusions: include more future works that could use this one as a starting point.
In references: review format.
Reviewer 2 Report
I can't do suggestion because this study is not relevant for clinical evaluation of the patients !
Reviewer 3 Report
Pediatric COVID-19 has a very rapid and often unnoticed clinical picture, but there is no published data in the literature on the correlation between PCR amplification cycles of SARS-COV-2 and disease severity in the age group studied from respiratory samples and gene amplification of SARS-COV-2 was correlated with disease severity as assessed by the LqSOFA score for the primary endpoint that was evaluated . Also as a secondary objective they determined if this score could predict the duration of symptom days of positive cases. It was observed that the duration of symptoms .Thus the LqSOFA score at the time of admission, can predict the duration of symptoms and the PCR amplification of SARS-COV-2 alone does not seem to play a key role in how the disease will progress in the age group studied. The role of CT as an adjunct to or replacement for reverse transcription– polymerase chain reaction (RT-PCR) in the screening or diagnosis of COVID-19 pneumonia has been the subject of much debate.
Revise!
It is necessary to revise the figures . Figure 2 is missing and should be included in the text. Also revise the statistical analysis (text). In page 5 the Figure 1 is duplicated. The numbering of the figures. Where is Figure 2 . Revisar e adequar com o texto.
Round 2
Reviewer 1 Report
Dear authors,
I congratulate you for your effort. This new version, in my humble view, is significantly better than the previous one. I only have two format considerations. Review the use of bold letter in table title and the final conclusion paragraph and the distance between lines in figure 2 title.
Reviewer 2 Report
Nothing to add !
Author Response
Dear Reviewer, thank you for your suggestions to revise our text to enrich and improve it.
Best Regards
C.A. Vincenzo Sortino